# Psychosocial Implications of Supportive Attitudes towards Intimate Partner Violence against Women throughout the Lifecycle

**DOI:** 10.3390/ijerph17176055

**Published:** 2020-08-20

**Authors:** Andrés Sánchez-Prada, Carmen Delgado-Alvarez, Esperanza Bosch-Fiol, Virginia Ferreiro-Basurto, Victoria A. Ferrer-Perez

**Affiliations:** 1Faculty of Psychology, Pontifical University of Salamanca, 37002 Salamanca, Spain; asanchezpr@upsa.es (A.S.-P.); mcdelgadoal@upsa.es (C.D.-A.); 2Faculty of Psychology, University of Balearic Islands, 07122 Palma, Spain; esperanza.bosch@uib.es (E.B.-F.); virginia.ferreiro@uib.es (V.F.-B.)

**Keywords:** intimate partner violence against women, attitudes, lifecycle, implicit association

## Abstract

Supportive attitudes towards intimate partner violence against women (IPVAW) normalize and promote these aggressive behaviors. As a result, more and more research is proposing the identification, analysis and intervention of these attitudes. However, the vast majority of this research focuses on students. The main objective of this paper is to analyze these supportive attitudes throughout the lifecycle. An opportunity sample of 200 Spanish participants, by age and sex fixed quotas, took part in this study. Attitudes were measured using the Inventory of Distorted Thoughts about Women and Violence, the Inventory of Beliefs about Intimate Partner Violence and the Gender Violence Implicit Association Test, a personalized form of Implicit Association Test (IAT). The results show that explicitly measured supportive attitudes towards intimate partner violence against women differ between age groups, adopting a U-shape distribution: lower acceptance among middle-aged-adults and young-adults and higher acceptance among adolescents and older adults. However, when these attitudes were implicitly measured, the IPVAW rejection increased with age, which is a counter-intuitive result and inconsistent with previous theoretical evidence. In summary, these results support an age effect that differs according to the measure of attitudes used and highlight some difficulties related to based-on-reaction-time measures among older people. This suggests the need for further research on the topic, especially among the older population.

## 1. Introduction 

Intimate partner violence against women (IPVAW) is “one of the most common forms of violence experienced by women” [1] (pp. 1527), and nowadays it is recognized by the majority of international (e.g., UN-Women, WHO, etc.) and national organizations (e.g., national governments including the Spanish Government) as a “global public health problem of epidemic proportions, requiring urgent action” [2] (pp. 3).

Research suggests that supportive attitudes towards IPVAW normalize and promote these aggressive behaviors [3,4,5,6,7,8,9,10]; therefore, more and more researchers consider the identification, analysis and intervention of these attitudes to be necessary [11,12]. These supportive attitudes towards IPVAW may be characterized as [4,12,13]: acceptability (i.e., approval, tolerance and permissively of this violence); minimization of their importance; and legitimation (i.e., normalization, victim blaming and justification or exoneration of the perpetrator). 

The available findings show that these attitudes are modulated by individual, organizational, community and social factors [4,14], including gender, age, educational level, political orientation, income level, marital status, place of residence, country of origin and experiences of victimization, in addition to the country’s level of development, its location on equal terms, its culture or its level of religiosity [15,16,17,18]. However, cross-cultural analyses show that there are regional and cultural differences in these attitudes [2,8,10,16,17,19,20,21,22]. 

In this paper we focus specifically on results and data from Europe, and in particular from Spain (with only some specific references to data from other countries), and on these supportive attitudes towards IPVAW throughout the lifecycle. 

Related to IPVAW in Spain, available data indicate that 11–25% of Spanish women have suffered IPVAW at some point in their life, and 3–15% have experienced it in the last year [23]. The measurement of explicit attitudes toward IPVAW demonstrates that around 90–95% of public opinion rejects this violence [24,25]. In this sense, it is important to remember that Spanish society has in the last few decades taken a process of change in public attitudes regarding what is (or is not) acceptable in intimate relationships, which may being understood as a reflection of social and cultural changes in norms concerning women behavior and IPVAW [25].

Related to the supportive attitudes towards IPVAW throughout the lifecycle, there is some evidence about an age effect on these attitudes, although research on the subject yields contradictory results [4,10,14].

To begin, the results of some studies indicate that the age effect on IPVAW acceptance could be considered negligible [19,26,27].

Additionally, some studies have observed a direct relationship between IPVAW acceptance and age, with older respondents being more likely to see women as provocative, accept violence as a normal and justifiable behavior and blame men’s lack of control and substance abuse for their aggressive behavior [13,28,29,30]. These results have been interpreted in terms of the cultural shift over time, so that people who reached adulthood before IPVAW became a social and public policy issue would be more inclined to hold traditional views about this violence.

Moreover, other studies have observed an inverse relationship between IPVAW acceptance and age, with this acceptance being greater among younger subjects [14,31,32]. These findings have been explained on the basis of three hypotheses: (i) these results are thought to be mediated by educational level and could reflect the educational differences between younger and older individuals, and the lack of experience, knowledge and exposure to the liberalizing influence of late secondary school and university education among younger people [4,10]; (ii) the results could be a reflection of the changes in attitudes, empathy, sensitivity or moral conscience that occur during adolescence and the transition to adulthood [4]; and (iii) these results may reflect the characteristics of the peer cultures among boys, taking into account that gender segregation and homophobia peak in early adolescence, and this context may be particularly prone to expressing views tolerant of IPVAW [4].

Finally, some research shows an agreement U-shaped pattern with these IPVAW supportive attitudes, with the highest levels of agreement at the beginning and at the end of the age range, and the lowest in the middle [33]. This type of curvilinear relationship has also been observed in some studies on sexist beliefs, with high levels of sexism among young people, lower levels in adulthood and again high levels in older people [34,35,36]. A possible explanation for these results could be that “the feminist movement of the 1960s changed attitudes in young people at that time, who are now middle-aged, and those gains have receded as the movement has become less salient and backlash has set in” [34] (p. 74). This change would not have affected older people in the same vein, as they received other cultural influences [25].

As different studies point out [4,10,14,29,37], such heterogeneous results may be caused by the fact that the age effect on IPVAW acceptance could be mediated by other factors, such as gender. In fact, there is ample evidence about a gender effect on attitudes towards IPVAW with consistent results indicating that in high-income countries (such as Western European countries, including Spain) men show more supportive attitudes than women [4,8,10,13,26,27].

It is worth mentioning that, despite the relevance of age factor, the vast majority of available research focuses on young people, and in particular on university students [9,12]. For example, Gracia, Santirso and Lila [12] carried out a systematic review of quantitative studies addressing attitudes towards IPVAW conducted in the EU, and they identified 62 empirical studies published from 2000 to 2018 that meet the fixed inclusion criteria. Of these studies, 45.2% involved students (19.4% high school students and 28.8% college students), and 48.4% were carried out in Spain, a country heavily involved in the fight against IPVAW. However, this is not a one-time event and other studies not included in this systematic review (because they were published subsequently or in the Spanish language and therefore did not meet the inclusion criteria) were also mostly conducted with young or university students [34,38,39]. 

It is also worth pointing out that the majority of research on attitudes towards IPVAW is carried out with explicit measures (such as self-reports), with an implied bias of social desirability [11,13,40] and the consequent limitations for the results obtained [41,42,43,44,45,46]. In fact, from the 62 empirical studies included in Gracia, Santirso and Lila’s [12] aforementioned systematic review, only one study used an implicit measure of attitudes toward IPVAW [13]. 

Previous research on attitudes using the IAT, a response time-based measure and one of the most widely used implicit measures, showed that older subjects tend to show larger IAT effects than younger subjects [47,48,49,50], especially when the original scoring algorithm [51] is used, and hypothesized that these results may be related to the effects of variations in cognitive fluency and speed of processing due to subject age, which would in turn produce variations on response latency. As Hummer et al. [48] pointed out: “IAT involves a simple (congruent) and a complex (incongruent) task, and task complexity is central to the logic behind the IAT: response latencies are expected to be faster to the congruent judgments than to the incongruent because the former are consistent with participants’ social schemas (i.e., simple) and the latter are not (i.e., complex)” [48](pp. 488). In other words, IAT effects are determined by general cognitive abilities, and older people generally show a larger increase in reaction time from simple to complex tasks than do younger people due to changes in cognitive abilities that usually decline with age [41,52,53]. Subsequent research has pointed out that an improved scoring algorithm, suggested by Greenwald, Nosek and Banaji [54], reduces this relationship between age and IAT scores [41,53]. Moreover, as Gonsalkolare et al. [55] suggested, an alternative interpretation could be that the older adults lacked motivation and were not engaged in the computer task required to complete an IAT measure.

Additionally, it is necessary to consider the possibility that apparent changes from using explicit attitudes measures may be not observed in implicit measures or vice versa, given that correlations between implicit and explicit measures tend to be low [56,57,58]. However, due to the scarcity of research carried out with these tools, it is difficult to know if an age effect on supportive attitudes towards IPVAW is produced or not when they are assessed by implicit measures.

In this context, the main objective of this paper is to analyze the supportive attitudes towards IPVAW throughout the lifecycle among Spanish people, by using both explicit and implicit measures of these attitudes and comparing the results among men and women in the different age groups. This allow us to compare attitudes towards IPVAW among different generations and their changes, questions which are often overlooked in studies whose samples are comprised mainly or exclusively of college students. 

## 2. Materials and Methods 

### 2.1. Participants

An opportunity sample of 200 Spanish participants, by age and sex fixed quotas, voluntarily took part in this study. Participants were recruited by announcements in different Spanish education centers (participating universities, vocational training centers and high schools). These volunteers had to be available to go to the university laboratory in the city where they lived and to spend at least 1 h participating in the study. Four age quotes were fixed: adolescents (<18 years), young adults (18–29 years), middle-aged adults (30–59 years) and older adults (≥60 years). Regarding the sex quota (50% male and 50% female), it must be pointed that it was not possible to recruit a sufficient number of male participants (*n* = 25) among middle-aged adults. Therefore, this sub-group was completed with female participants up to the quota of 50 participants per group. Table 1 shows the characteristics of the recruited sample.

### 2.2. Instruments

The Inventory of Distorted Thoughts about Women and Violence, (Spanish abbreviation: IPDMV) [59] (adapted version of Ferrer et al. [60]) assesses four attitude dimensions: Inferiority of Women Compared to Men (F1-IW, 7 items; Cronbach’s *α* = 0.86); Blaming Female Victims of Abuse (F2-BW, 7 items; Cronbach’s *α* = 0.62); Violence as an Appropriate Problem-solving Strategy (F3-VP, 5 items; Cronbach’s *α* = 0.69); and Minimization and Exoneration of the Abuser (F4-MA, 5 items; Cronbach’s *α* = 0.53). Responses are given on a four-point scale from 1 (totally disagree) to 4 (totally agree), whereby the higher are the scores, the higher are the levels of distorted thoughts (i.e., lower scores indicate IPVAW rejection). 

The Inventory of Beliefs about Intimate Partner Violence (IBIPV) [61] (revised model of Sanchez-Prada et al. [62]). This instrument was constructed on the basis of Inventory of Beliefs about Wife Beating (IBWB [63]) and assesses three dimensions on a seven-point scale from 1 (totally disagree) to 7 (totally agree): Justifying Partner Violence (F1-JPV, 5 items; Cronbach’s *α* = 0.64); Victims Responsible for Violence (F2-VRV, 8-ítems; Cronbach’s *α* = 0.86); and Abuser Responsible for Violence (F3-ARV, 5 items; *α* = 0.93). After reversing the item scores of dimension F3-ARV, higher scores indicate higher tolerance towards IPVAW (i.e., lower rejection). 

The Gender Violence Implicit Association Test (GV-IAT [64,65,66]) is a form of personalized IAT used as an implicit measure of attitudes towards IPVAW with two target categories—Gender Violence vs. Non-Gender Violence—and two attribute categories—Good vs. Bad. Related to the target category, it is important to note that, under Spanish law [67], IPVAW is known as gender violence (see [25]), which is why GV-IAT uses the term “Gender Violence” to refer to IPVAW. The fundamental principle of IAT is that, where two concepts are strongly associated (compatible phase), the response latency (RL) is less than when this is not the case (incompatible phase). The IAT effect (D-scores) is calculated using the difference between incompatible-compatible critical phase latency-based responses [54,64]. The typical IAT procedure with feedback was used [64,66], and the participants completed the GV-IAT task in seven blocks, of which three were considered practice trials (B1–B2 including 24 trials and B5 including 48 trials) and four as critical blocks (B3–B6 including 24 trials in the compatible phase and B4–B7 including 48 trials in the incompatible phase) [54]. Additionally, the first trials of each critical block were considered “practice trials” and only the next trials of each block were computed in the D score (in the end, 20 trials were computed in B3–B6 and 40 trials in B4–B7). In addition, it is important to note that we use the procedure “built-in penalty” in error latencies by computing the accumulated time of the wrong-plus-corrected response [54,68,69]. In summary, the higher is the D-score value, the higher is the difference between response latencies in incompatible versus compatible phases and, therefore, the stronger is the implicit rejection of IPVAW. The internal consistency by the split-half reliability method [54] recommended by Kurdi et al. [70] yielded an average estimate of 0.73 [64], indicating an adequate reliability [49,71].

Finally, we administered an ad-hoc questionnaire including questions about sociodemographic data (age and gender, by responding the question “Tell me what gender you identify with?”), and about the perceived frequency and severity of different forms of IPVAW (from 1 = nothing serious to 4 = very serious): physical violence, psychological violence and sexual violence. When asking about IPVAW perceived severity, we used exactly the same questions of the 2010 Eurobarometer [24] for this aim. In addition, we included two specific questions about perception of protective measures for victims and laws to punish abusers (from 1 = very insufficient to 4 = very adequate).

### 2.3. Procedure

Experimental sessions took place in the labs at each participating university. Upon arrival, participants had to read over a brief description of the study and the corresponding consent form. Upon providing their informed consent, participants were asked to perform the IAT task and then complete the IPDMV and IBIPV. Finally, they responded to the ad-hoc questions on sociodemographic data and IPVAW-related perceptions. The Bioethics Committee of one of the participating universities approved the study.

### 2.4. Data analysis

The SPSS 25.0 (IBM Corporation, Armonk, NY, USA) statistical program was used for data analysis. A descriptive study was performed to report the characteristics of the sample subjects including means and standard deviations and frequency distributions. The assumption of normality was not met for any of the explicit measures of IPVAW-attitudes (Kolmogorov–Smirnov test *p* < 0.001 for all measures). Consequently, various non-parametric tests were carried out: Kruskal–Wallis tests to compare the attitude scores among age groups and Mann–Whitney U tests to compare the scores by gender. Pearson’s indices were used to estimate the correlation between measures. Under the assumption of normality (Kolmogorov–Smirnov test: 0.54, *p* = 0.200; *n* > 30) and homoscedasticity (Levene’s test: *F* (3, 196) = 1.493; *p* = 0.218) of the GV-IAT measure, parametric tests were used to compare D-score differences between age groups. Specifically, an Analysis of Variance (ANOVA) and an Analysis of Covariance (ANCOVA) introducing latency-response (LR) as covariate were carried out to compare implicit measures (GV-IAT) between age groups. Given that the assumption of normality was not met for answers related to IPVAW perceived frequency and severity, nor for measures related to protect victims and punish abusers (Kolmogorov–Smirnov test *p* < 0.001 for all measures), non-parametric Kruskal–Wallis tests were used for comparison purposes. Through multiple post-hoc comparisons between groups, homogeneous sets were obtained: groups whose difference is not significant were assigned to the same set, and groups whose difference is significant were assigned to different sets. Finally, non-parametric U Mann–Whitney tests were carried out to compare male and female scores in explicit and implicit attitudes towards IPVAW, in each of the age groups.

## 3. Results

### 3.1. Explicit Measures of Attitude towards IPVAW

By comparing the explicit attitudes towards IPVAW among age groups, statistically significant differences were found in all IPDMV dimensions (Table 2). Multiple comparisons between groups yielded two homogeneous sets for all dimensions: *lower acceptance* [Set 1] and *higher acceptance* [Set 2]. Adolescents (<18) and older adults (≥60) performed in *higher acceptance,* whereas middle-aged adults (30–59) and young adults (18–29) performed in *lower acceptance*, except in F3-VP, where young adults performed in the *higher acceptance* set with adolescents and older adults.

Figure 1 shows how attitudes differ between age groups by adopting a U-shape.

The Kruskal–Wallis test carried out with IBIPV scores yielded similar results (Table 3). That is, statistically significant differences were found in all IBIPV dimensions, and multiple comparisons among groups also yielded two homogeneous sets for all dimensions: *lower acceptance* [Set 1] including middle-aged adults and young adults and *higher acceptance* [Set 2] including adolescents and older adults. In the F3-ARV dimension, statistically significant differences were obtained only between young adults [Set 1] and adolescents [Set 2], with the other groups having intermediate scores, without significant differences between them. 

Figure 2 shows that the means by age groups in IBIPV dimensions adopted the same U-shape as in the IPDVM dimensions. 

### 3.2. Implicit Measures of Attitude towards IPVAW

By comparing the implicit attitudes towards IPVAW among age groups, significant differences were found between age groups. As shown in Table 4, D-scores increase linearly with age, which would indicate that the implicit rejection of IPVAW increases with age, in contrast to the evidence available for the explicit measures. Previous evidence suggests that the analysis and interpretation of IAT measures must consider age-related slowing and that the IAT results reflect not only implicit perceptions but also age group differences in speed of processing [48]. Considering this evidence, we analyzed the age effect on D-scores calculating the Pearson’s correlation between these variables. A correlation value of 0.364 (*p* < 0.001) was obtained, confirming the existence of an effect of cognitive variables linked to age on the execution of the IAT.

To further investigate this outcome, we made a Pearson’s correlation between age and latency responses (*r* = 0.420, *p* < 0.001 in phase compatible, and *r* = 0.576, *p* < 0.001 in phase incompatible). This result confirms a large effect of age on the latency response in our sample. In fact, “it is expected that IAT effects will be artifactually larger for any subjects who respond slowly, not just the elderly (…). It is desirable for an IAT measure to minimize this undesired artifactual correlation with response speed” [54] (pp. 200). To control this undesired GV-IAT effect, we carried out a covariance analysis (ANCOVA), controlling the effect of the latency responses on the D scores and introducing the latency of the incompatible phase (LRi) as a covariate, since it shows a greater relationship with age. Table 4 shows these results and the two homogeneous sets from post-hoc comparisons using the Tukey test.

As shown in Figure 3, controlling the effect of response latency minimizes the differences in IAT effects among age groups; however, a linear increasing of D scores with age continues to persist. 

Adopting the cut-off points proposed by Greenwald et al. [54] for interpreting D-scores, four IPVAW rejection intervals could be established from the GV-IAT scores: null rejection (D < 0.20), mild rejection (0.20 ≤ D < 0.50), moderate rejection (0.50 ≤ D < 0.80) and strong rejection (D ≥ 0.80). As shown in Figure 3, if the GV-IAT effect only measured attitude, these results would indicate that the older adults group would have a strong implicit rejection of IPVAW, higher than the other three age groups (which would have a mild level of rejection). Moreover, the results obtained (in ANCOVA analysis) show that this difference is not attributable to age and response latency (RL) correlations, since the differences remain unchanged after eliminating the influence of LR on D-scores.

### 3.3. Other Evidence about Attitudes towards IPVAW 

Other attitude-related indicators, such as questions included in studies about perceived frequency and severity of different types of IPVAW, were analyzed in this study. Specifically, we asked participants about the perceived frequency and severity of different forms of IPVAW (from 1 = nothing serious to 4 = very serious): physical violence; psychological violence; sexual violence and about perception of protective measures for victims and laws to punish abusers (from 1 = very insufficient to 4 = very adequate).

Table 5 shows the results related to IPVAW perceived frequency.

Here again, two homogeneous sets are identified for the variables whose differences are statistically significant: *lower acceptance* [Set 1, i.e., *higher perceived frequency*] and *higher acceptance* [Set 2, i.e., *lower perceived frequency*]. Regarding perceived frequency of psychological abuse, attitudes differ between age groups by adopting a U-shape distribution: adolescents (<18) and older adults (≥60) performed in Set 2, whereas young adults (18–29) and middle-aged adults (30–59) performed in Set 1. In the case of violent threats, only older adults performed in Set 2 whereas all the other groups performed in Set 1. Finally, for perceived frequency of freedom restrictions, adolescents sit between the two sets. Figure 4 shows these differences.

Differences in the perceived severity of different types of IPVAW were statistically significant for all variables, as shown in Table 6.

In general, a similar pattern was observed for perceived severity of different IPVAW, as described above, adopting a U-shape distribution with young adults and middle-aged adults showing *lower acceptance* [Set 1, i.e., *higher perceived severity*] and older adults and adolescents showing *higher acceptance* [Set 2, i.e., *lower perceived severity*]. In this particular case, a third set [Set 3] appears for perceived psychological abuse, indicating more perceived severity differences between the age groups. In this case, the most extreme differences are observed between middle-aged adults [Set 1, *lower acceptance*] and older adults [Set 3, *higher acceptance*]. A particularly interesting result is that the only type of IPVAW that adolescents perceive as “more severe” is sexual abuse, more in line with the results obtained from young adults and middle-aged adults and further from those of older adults. Figure 5 shows these results.

The last questions referred to the participants’ perception about measures for protecting victims and punishing abusers. Results obtained yielded statistically significant differences among age groups, as shown in Table 7. Once again, two sets were observed: *lower acceptance* [Set 1, i.e., *lower sufficiency of measures*] and *higher acceptance* [Set 2, i.e., *higher sufficiency of measures*]. Young adults and middle-aged adults performed in Set 1, whereas adolescents and older adults performed in Set 2. However, regarding the sufficiency of laws to punish batterers, no significant differences were found between middle-aged adults (30–59), who had intermediate scores, and other groups.

Figure 6 shows the U-shape distribution adopted by these variables.

### 3.4. Gender Differences

When comparing male and female scores of explicit and implicit attitudes towards IPVAW into each of the age groups, no statistically significant differences were found among groups in any case, although the scores among men were greater in explicit acceptance and lower in implicit IPVAW acceptance. To deepen the analysis, we regrouped the four age groups into two age groups to increase the *n* of samples: 16–29 years (adolescents and young adults) and ≥30 years (middle-aged adults and older adults). For the ≥30 years age group, the analysis did not yield any statistically significant differences between men and women (α = 0.05). The results for the ≤29 years age group can be seen in Table 8, which show significant differences between women and men in all dimensions of attitudes towards IPVAW measured explicitly.

Figure 7 shows the means of men and women in the ≤29 years group. The greatest differences between men and women were yielded in dimensions related to the exoneration of the abuser (F3-ARV and F4-MA).

## 4. Discussion

The study carried out has enabled us to achieve the main objective proposed: to analyze supportive attitudes towards IPVAW throughout the life cycle. The results obtained indicate that, as suggested by the literature on the issue [4,10,14], there is an age effect on these attitudes. In fact, the results obtained show statistically significant differences between different age groups for all dimensions of attitudes towards IPVAW, whether those attitudes are assessed with explicit or implicit measures.

However, it is important to note that this effect manifests itself differently depending on the type of attitude assessment tool used, as in previous research about attitudes [50,56,57,58]. Specifically, the age effect in our research for supportive attitudes towards IPVAW measured explicitly adopted a U-shape distribution, with lower acceptance among middle-aged adults and young adults and higher acceptance among adolescents and older adults. In general, the results obtained for the other attitude-related indicators analyzed, such as perceived frequency or severity of IPVAW types, confirm that attitudes differ between age groups, adopting a U-shape distribution in the same sense aforementioned. These U-shape results are similar to other results obtained in both supportive IPVAW attitudes [33] and sexism [34,35,36]. 

In relation to these results, the hypotheses used to explain the inverse age effects on attitudes towards IPVAW may be useful to understand the results we obtained among the youngest participants. The high levels of IPVAW acceptance in this age group could be mediated by educational level and reflect the lack of experience, knowledge and exposure to the liberalizing influence of late secondary school and university education among these individuals; they could be a reflection of the changes in attitudes, empathy, sensitivity or moral conscience that occur during adolescence and the transition to adulthood; or they may reflect some characteristics of the peer culture among boys, by the affirmation of traditional masculinity through sexist or homophobic attitudes [4]. It should be noted that the Spanish education system has in recent decades included some content related to the equality between women and men and awareness of IPVAW [72]. However, these results indicate that the effect of these actions is still limited, and consequently further hard work in this direction is needed to improve awareness and sensitivity to this violence.

Furthermore, as Garigordobil [34] pointed out, results from the adults and older adults could be due to the different cultural influences received by different generations and, especially, to the influence of the feminist movement on people who are now adults. In fact, even in the unfavorable environment of the Franco regime (1939–1975), from the 1960s onwards, the Spanish feminist movement began to take shape and, after the death of Franco and in parallel with the process of political transition towards democracy, began to accomplish certain achievements (the legalization of contraceptives in 1977, the inclusion of the principle of equality before the law in the Constitution of 1978, the legalization of divorce in 1981, etc.) which led to important changes in the roles of Spanish women [73]. One could assume that these processes have had an effect on the generations that lived through them and who are now adult and middle-aged [34]. In contrast, older Spanish people were socialized in a very different context. In this sense, it could be noted that Franco’s regime developed an important ideological control of the whole of society which not only intended to maintain the political regime, but also to enshrine the patriarchal sexual division of work, with women confined to the private world, performing the role of wife and mother, and men focused on the public world, playing the role of providers, creating a context where IPVAW did not “exist” (was not categorized as a crime and was not recognized as a problem) and, therefore, was neither visible nor quantifiable. Comprehending this circumstance helps to understand the starting point of the Spanish society to deal with this issue and the probable lack of social awareness among some older people [25]. On the other hand, the recent emergence of the extreme right into Spanish politics, and its popularity among younger people, could influence the greater acceptance of IPVAW in the group of adolescents. Further research is needed to explore in greater depth the effect of IPVAW denial among this political group and on increasing IPVAW acceptance among adolescents.

When we assessed attitudes towards IPVAW with implicit measure tools, a direct relation between age and attitudes was obtained, showing that the IPVAW rejection increased with age (note that the higher is the D-score, the greater is the IPVAW rejection). It should be pointed out that the correlations between age and D-score and between age and latency responses provide evidence of the existence of a large effect of cognitive variables linked to age on the IAT execution (in this case, on the GV-IAT execution). To control this effect, we carried out a covariance analysis (controlling the effect of the latency-responses on the D scores, and introducing the latency of the incompatible phase (LRi) as a covariate). Although this test minimizes the differences in IAT effects among age groups, a linear increasing of D scores with age persists, so that the older adults group shows a strong implicit IPVAW rejection, higher than the other younger people. This is a counter-intuitive result, and it is inconsistent with the aforementioned previous theoretical and empirical evidence. To explain these contradictory results, we can consider several hypotheses.

To begin, it is important to take into consideration the probable effect of cognitive factors. As mentioned above, IAT effects are determined by general cognitive abilities and, in general, older people show a larger increase in reaction time and, consequently, larger IAT effects than younger people [41,48,52,53]. Although this hypothesis may not be completely ruled out, the use of the statistical analyses described above, as well as the use of the improved scoring algorithm [54] that would reduce the relationship between age and IAT scores [41,53], suggest that the longer latency times, which are expected in older people, influence, but are not a sufficient explanation for, these results. It is therefore essential to continue carrying out studies that help us to study in depth the effect of cognitive variables on the IAT effect, as well as on the GV-IAT effect.

Additionally, as Gonsalkolare et al. [55] suggested, an alternative interpretation to the effect of cognitive factors could be that older adults lacked motivation and were not engaged in the computer task required to complete an IAT measure. In this regard, it is important to note that one of the most important difficulties in the recruiting process of our sample was in fact the reluctance of some older people to perform computer-based tasks. It could be hypothesized that this reluctance may have had some effect on the results obtained; it is therefore important to insist that more studies are needed that include samples of people of all ages, not just youths or students.

Moreover, it has been hypothesized that the age effect on IPVAW acceptance may be mediated by factors such as gender [4,10,14,29,37]. The analyses performed in this study showed that, although scores among men were greater in explicit acceptance and lower in implicit IPVAW acceptance, no statistically significant differences were found among age groups in any case; and when two age groups were re-grouped (adolescents and young adults on the one hand and middle-aged adults and older adults, on the other hand), only among the youngest people were there significant differences between women and men in attitudes towards IPVAW measured explicitly. These differences occur in the same sense as described in the previous literature on the subject, i.e., men show more IPVAW supportive attitudes than women [4,8,10,13,26,27]. Again, only the results obtained through explicit measures of attitudes are in accordance with the literature on the topic, although the small size of the samples studied (which requires a larger mean difference between groups to obtain significant results) may be the reason no differences between men and women were observed in all age groups.

Finally, it is also worth pointing out that correlations between implicit and explicit measures tend to be low [57,58], and the discrepancy between implicit and explicit measures is more often the standard than the exception [50]. This evidence underscores the need for greater empirical support to advance towards the construction of a theoretical interpretative framework for the meaning of IAT measures, particularly in cases such as IPVAW where these types of measure tools have been infra-utilized up until now.

The main strength of this paper is to include people from a wide range of age groups, extending beyond the nearly exclusive inclusion of young people and students, as with other research on the issue [9,12]. However, in our view, although this study makes some contributions to the field, it is not without limitations, one of the most important being the characteristics of the sample, i.e., an incidental sample, with a limited size and a limited number of participants in the different age groups studied. Another issue that could constitute a limitation has to do with the measures used. Thus, beyond the difficulties of the implicit measures aforementioned, it is possible that the explicit measures used may have become outdated or are too explicit in what they ask. Note that IPDMV was developed in the 1990s specifically for clinical issues [59,60] and IBIPV is based on a previous tool from the 1980s [61]. Since then, social sensitivity towards IPVAW as well as the Spanish legislation itself have been substantially modified [25]. Another limitation of this paper is related to the factors considered. In this sense, the main objective of this paper was to analyze IPVAW supportive attitudes throughout the lifecycle and consequently the analysis performed were centered in identifying differences between age groups. However, some other analysis may be performed, such as studying intra-group gender differences or also included socio-cultural differences. Finally, an additional limitation has to do with the cross-sectional nature of the study, and with the impossibility of speaking in evolutionary terms given this nature. Therefore, it would be necessary to develop longitudinal studies in each cohort, allowing a follow-up and an analysis of the evolution and changes in IPVAW attitudes with age and other related factors.

## 5. Conclusions

This study allowed us to investigate questions related to generational attitudes towards IPVAW, which are often overlooked in studies whose samples are primarily or exclusively comprised of college students. Our results show that this effect of age exists, although it manifests itself in different ways depending on whether these attitudes are measured explicitly (U-shaped effect) or implicitly (direct relationship).

These results have important practical and methodological implications. On the one hand, from a practical point of view, the results obtained by applying explicit measures of attitudes towards IPVAW might point to: (a) the benefits of education and social mobilizations on equality between women and men and against IPVAW, which reflect on the levels of IPVAW rejection shown by adults and middle-aged people; and (b) the effects of the lack of these factors, which reflect on the acceptance levels of IPVAW shown by older people who did not have these influences. Furthermore, these same results reaffirm the need to continue working on the education of younger people to counteract some of the other influence factors aforementioned [34] and avoid the high rates of IPVAW found in these populations [23]. On the other hand, from a methodological point of view, the results obtained by applying implicit measures of attitudes towards IPVAW (and some obtained by applying explicit measures) help us to delve deeper into determining what the advantages and disadvantages of the different tools may be in general, as well as in the different age groups.

## Figures and Tables

**Figure 1 ijerph-17-06055-f001:**
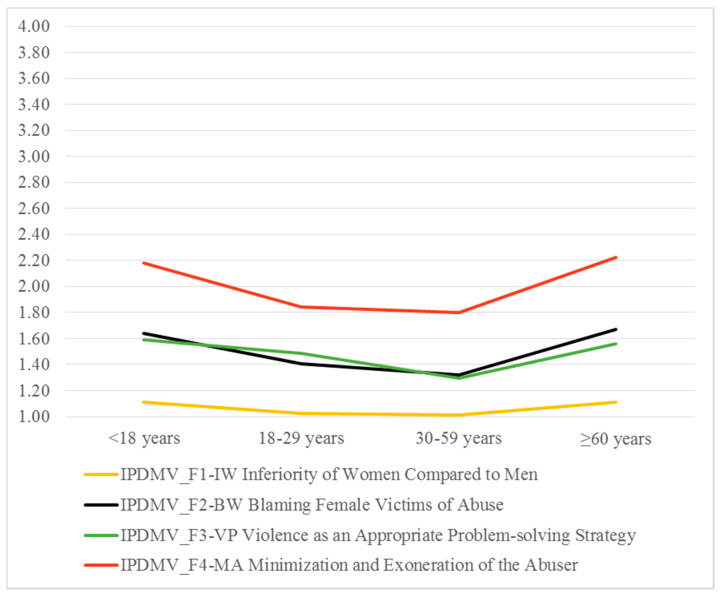
IPDMV: Means by age groups.

**Figure 2 ijerph-17-06055-f002:**
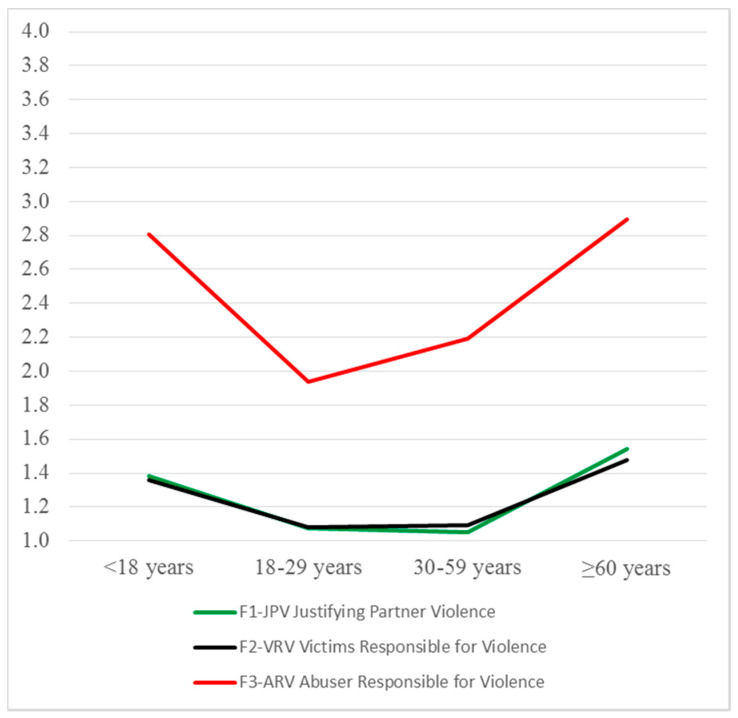
IBIPV: Means by age groups.

**Figure 3 ijerph-17-06055-f003:**
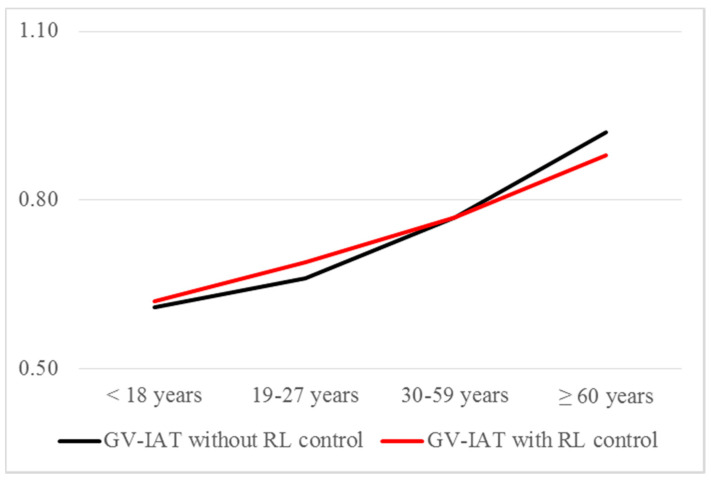
GV-IAT effects with and without RL control.

**Figure 4 ijerph-17-06055-f004:**
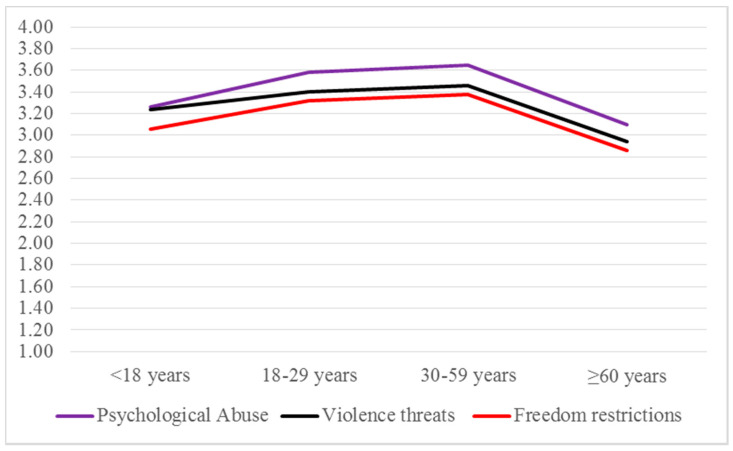
Perceived frequency of IPVAW types: means by age groups.

**Figure 5 ijerph-17-06055-f005:**
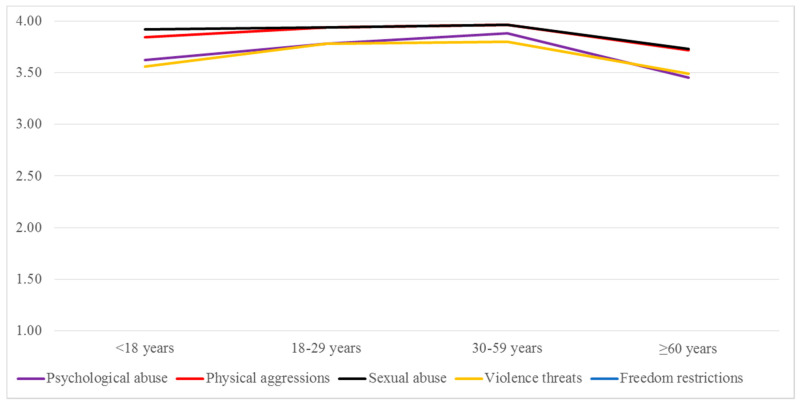
Perceived severity of IPVAW types: means by age groups.

**Figure 6 ijerph-17-06055-f006:**
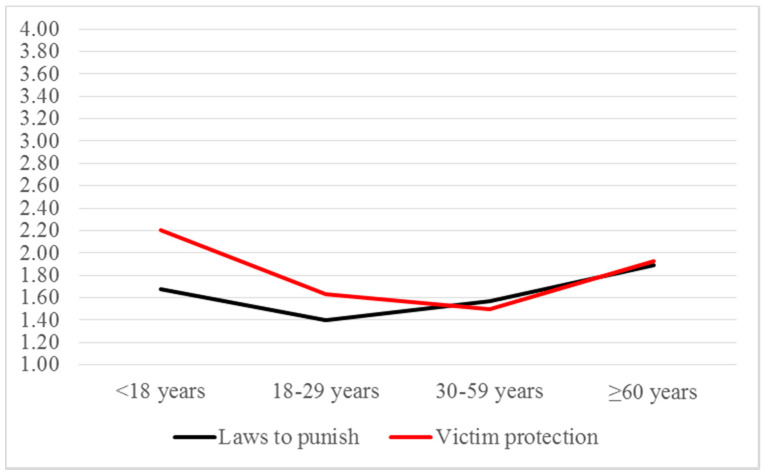
Perception about measures for protecting victims and punishing abusers: means by age groups.

**Figure 7 ijerph-17-06055-f007:**
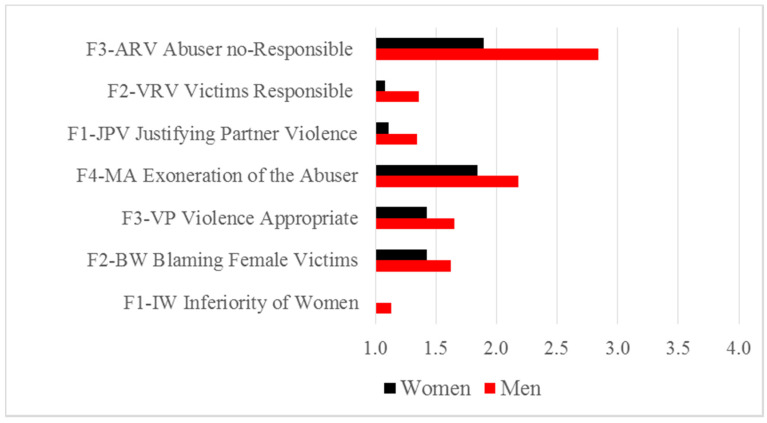
Means in explicit measures of attitudes toward IPVAW by gender (≤29 years).

**Table 1 ijerph-17-06055-t001:** Participant’s characteristics.

		<18 Years (*n* = 50)	18–29 Years (*n* = 50)	30–59 years (*n* = 50)	≥60 Years (*n* = 50)
Age	Mean *SD* Range	16.60 0.50 16–17	19.82 2.30 18–27	41.64 9.99 30–59	68.58 5.30 60–82
Gender	Men Women	25 (50%) 25 (50%)	25 (50%) 25 (50%)	16 (32%) 34 (68%)	25 (50%) 25 (50%)

**Table 2 ijerph-17-06055-t002:** IPDMV: Mean differences among age groups.

IPDMV Dimensions	Group	Mean	*SD*	χ^2^ (3 *df*)	*p*	Homogeneous Sets ^1^ Set 1 Set 2	
F1-IW: *Inferiority of* *Women Compared to Men*	<18	1.11	0.29	18.717	<0.001		<18
	18–29	1.02	0.08			18–29	
	30–59	1.01	0.05			30–59	
	≥ 60	1.11	0.19				≥60
F2-BW: *Blaming Female* *Victims of Abuse*	<18	1.64	0.41	26.873	<0.001		<18
	18–29	1.41	0.31			18–29	
	30–59	1.32	0.25			30–59	
	≥60	1.67	0.45				≥60
F3-VP: *Violence as an* *Appropriate Problem-solving* *Strategy*	<18	1.59	0.51	12.720	0.005		<18
	18–29	1.49	0.56				18–29
	30–59	1.30	0.40			30–59	
	≥60	1.56	0.51				≥60
F4-MA: *Minimization and* *Exoneration of the Abuser*	<18	2.18	0.67	16.137	<0.001		<18
	18–29	1.84	0.53			18–29	
	30–59	1.80	0.64			30–59	
	≥60	2.22	0.71				≥60

^1^ Set 1 = lower acceptance of IPVAW; Set 2 = higher acceptance of IPVAW.

**Table 3 ijerph-17-06055-t003:** IBIPV: Mean differences among age groups.

IBIPV Dimensions	Group	Mean	*SD*	χ^2^ (3*df*)	*p*	Homogeneous Sets ^1^ Set 1 Set 2	
F1-JPV: *Justifying* *Partner Violence*	<18	1.38	0.62	31.411	<0.001		<18
	18–29	1.08	0.25			18–29	
	30–59	1.05	0.18			30–59	
	≥60	1.54	0.98				≥60
F2-VRV: *Victim* *Responsible for Violence*	<18	1.36	0.72	15.883	0.001		<18
	18–29	1.08	0.20			18–29	
	30–59	1.09	0.25			30–59	
	≥60	1.48	1.14				≥60
F3-ARV: *Abuser* *Responsible for Violence*	<18	2.80	1.51	11.965	0.008		<18
	18–29	1.94	1.54			18–29	
	30–59	2.19	1.25			30–59	30–59
	≥60	2.90	2.31			≥60	≥60

^1^ Set 1 = lower acceptance of IPVAW; Set 2 = higher acceptance of IPVAW.

**Table 4 ijerph-17-06055-t004:** D-scores in GV-IAT: ANOVA and ANCOVA differences between age groups.

	**Group**	**Estimated Mean (*SD*)**	***F* (3, 196)**	***p***	**η^2^**	**1-β**	**Homogeneous Sets ^1^** **Set 1 Set 2**	
ANOVA	<18	0.61 (0.045)	9.507	<0.001	0.127	0.997		<18
	18–29	0.66 (0.045)						18–29
	30–59	0.77 (0.045)					30–59	30–59
	≥60	0.92 (0.045)					≥60	
	**Group**	**Estimated Mean (*SD*)**	**F (3, 195)**	***p***	**η^2^**	**1-β**	**Homogeneous Sets ^1^** **Set 1 Set 2**	
ANCOVA	<18	0.62 (0.046)	4.413	0.005	0.064	0.870		<18
	18–29	0.69 (0.047)					18–29	18–29
	30–59	0.77 (0.045)					30–59	30–59
	≥60	0.88 (0.053)					≥60	
Covariate RLi (Value 2121.88)			2.316	0.130	0.012	0.328		

^1^ Set 1 = lower acceptance of IPVAW; Set 2 = higher acceptance of IPVAW. ANOVA = Analysis of Variance. ANCOVA = Analysis of Covariance.

**Table 5 ijerph-17-06055-t005:** Perceived frequency of IPVAW types: mean differences among age groups.

Frequency	Group	Mean	*SD*	χ^2^ (3*df*)	*p*	Homogeneous Sets ^1^ Set 1 Set 2	
Psychological abuse	<18	3.26	0.67	25.644	<0.001		<18
	18–29	3.58	0.54			18–29	
	30–59	3.65	0.52			30–59	
	≥60	3.10	0.65				≥60
Physical aggression	<18	3.16	0.65	7.224	0.065		
	18–29	3.12	0.58				
	30–59	3.26	0.77				
	≥60	2.90	0.72				
Sexual abuse	<18	3.08	0.72	3.800	0.284		
	18–29	3.22	0.82				
	30–59	3.16	0.77				
	≥60	2.94	0.81				
Violence threats	<18	3.24	0.56	16.137	0.001	<18	
	18–29	3.40	0.67			18–29	
	30–59	3.46	0.65			30–59	
	≥60	2.94	0.63				≥60
Freedom restrictions	<18	3.06	0.84	18.801	<0.001	<18	<18
	18–29	3.32	0.74			18–29	
	30–59	3.38	0.78			30–59	
	≥60	2.86	0.74				≥60

^1^ Set 1 = lower acceptance of IPVAW; Set 2 = higher acceptance of IPVAW.

**Table 6 ijerph-17-06055-t006:** Perceived severity of IPVAW types: mean differences among age groups.

Severity	Group	Mean	*SD*	χ^2^ (3 *df*)	*p*	Homogeneous Sets ^1^ Set 1 Set 2 Set 3 ^2^		
Psychological abuse	<18	3.62	0.57	18.801	<0.001		<18	<18
	18–29	3.78	0.42				18–29	18–29
	30–59	3.88	0.33			30–59		
	≥60	3.45	0.61					≥60
Physical aggressions	<18	3.84	0.37	13.527	0.004	<18	<18	
	18–29	3.94	0.24			18–29		
	30–59	3.96	0.20			30–59		
	≥60	3.72	0.50				≥60	
Sexual abuse	<18	3.92	0.34	14.838	0.002	<18		
	18–29	3.94	0.24			18–29		
	30–59	3.96	0.20			30–59		
	≥60	3.73	0.49				≥60	
Violence threats	<18	3.56	0.58	15.844	0.001		<18	
	18–29	3.78	0.47			18–29		
	30–59	3.80	0.45			30–59		
	≥60	3.49	0.50				≥60	
Freedom restrictions	<18	3.58	0.64	9.683	0.021		<18	
	18–29	3.78	0.47			18–29	18–29	
	30–59	3.84	0.37			30–59		
	≥60	3.57	0.58				≥60	

^1^ Set 1 = lower acceptance of IPVAW; Set 2 = higher acceptance of IPVAW. ^2^ Psychological abuse: three acceptance sets were found: 1 (lower), 2 (intermediate) and 3 (higher).

**Table 7 ijerph-17-06055-t007:** Perception about measures for protecting victims and punishing abusers: mean differences among age groups.

	Group	Mean	*SD*	χ^2^ (3*df*)	*p*	Homogeneous Sets ^1^ Set 1 Set 2	
Laws to punish	<18	1.68	0.56	11.601	0.009		<18
	18–29	1.40	0.79			18–29	
	30–59	1.57	0.69			30–59	30–59
	≥60	1.89	0.96				≥60
Victim protection	<18	1.40	0.79	18.706	<0.001		<18
	18–29	1.63	0.86			18–29	
	30–59	1.50	0.55			30–59	
	≥60	1.93	0.81				≥60

^1^ Set 1 = lower acceptance of IPVAW; Set 2 = higher acceptance of IPVAW.

**Table 8 ijerph-17-06055-t008:** Mean differences in explicit and implicit measures by gender (≤29 years).

Explicit (IPDMV and IBIPV Dimensions) and Implicit Measures	Gender	Mean	*SD*	Z	*p*
F1-IW: *Inferiority of Women Compared to Men*	Men Woman	1.130 1.003	0.294 0.020	−4.165	<0.001
F2-BW: *Blaming Female Victims of Abuse*	Men Woman	1.620 1.426	0.404 0.332	−2.470	0.013
F3-VP *Violence Appropriate Problem-solving Strategy*	Men Woman	1.652 1.424	0.577 0.474	−2.165	0.030
F4-MA *Minimization and Exoneration of the Abuser*	Men Woman	2.180 1.840	0.631 0.581	−2.842	0.004
F1-JPV *Justifying Partner Violence*	Men Woman	1.348 1.112	0.564 0.375	−3.088	0.002
F2-VRV *Victims Responsible for Violence*	Men Woman	1.358 1.082	0.716 0.200	−2.717	0.007
F3-ARV *Abuser Responsible for Violence*	Men Woman	2.844 1.896	1.787 1.174	−2.777	0.005
GV-IAT *Implicit Rejection of Gender Violence*	Men Woman	0.619 0.652	0.287 0.285	−0.789	0.430

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
