# Peer review of "Psychosocial Implications of Supportive Attitudes towards Intimate Partner Violence against Women throughout the Lifecycle"

_ijerph, 2020, doi:10.3390/ijerph17176055_

Round 1
Reviewer 1 Report
This paper presents an important contribution to the literature regarding IPV - a life-cycle analysis. I have some minor comments to improve the manuscript:
- Life-cycle: be consistent throughout the manuscript regarding hyphenation. Why is it capitalized in the title?
- Abstract: Add (IPVAW) after it is spelled out so when you use the acronym later in the abstract it has been defined.
- Keywords: consider adding 'implicit association'
- Introduction:Page 2 lines 45-57 - this is broken up strangely. Make this a single paragraph or more clearly organize this.
- Introduction: page 2 lines 75 it can be confusing what is considered "supportive attitudes" add IPVAW before supportive to make this more clear.
- Introduction: Page 3 line 128 - "This will allow us to study questions related to..." Make these questions explicit.
- Instruments: Page 4 lines 143-157 - are these alpha values from prior literature or from data collected for this study?
- Procedure: Page 5 line 190 add "to" they responded to...
- Results: In the tables F1-IW, F2-BW - make clear what those codes are.
- Results: Check for commas vs periods in numbers. The y axes on most of the graphs have commas and they should be periods. There are commas in some of the chi square values as well.
- Results: 3.3 Other evidence section. Remind the reader here about this scale. I had to go back and look for what was being measured and on what response scale.
- Results 3.4 Gender differences: I thought you recruited by sex not gender... be clear and consistent about this throughout the manuscript.
Author Response
Responses to Reviewer 1:
Thank you very much for your assessment. We have made all the changes suggested by you.
1. Life-cycle: be consistent throughout the manuscript regarding hyphenation. Why is it capitalized in the title?
Author’s response: We have changed the title to be consistent (see line 4)
2. Abstract: Add (IPVAW) after it is spelled out so when you use the acronym later in the abstract it has been defined.
Author’s response: We have added IPVAW in the abstract (see line 14).
3. Keywords: consider adding 'implicit association'
Author’s response: We have added “implicit association” as a key word (see line 31).
4. Introduction: Page 2 lines 45-57 - this is broken up strangely. Make this a single paragraph or more clearly organize this.
Author’s response: We have rephrased the paragraph in the sense pointed out by the Reviewer (see lines 51-53 and 61-63).
5. Introduction: page 2 lines 75 it can be confusing what is considered "supportive attitudes" add IPVAW before supportive to make this more clear.
Author’s response: We have added IPVAW before “supportive” (see line 83).
6. Introduction: Page 3 line 128 - "This will allow us to study questions related to..." Make these questions explicit.
Author’s response: We have rephrased the sentence in the sense pointed out by the Reviewer (see lines 136-138).
7. Instruments: Page 4 lines 143-157 - are these alpha values from prior literature or from data collected for this study?
Author’s response: The alphas included in the text are values from prior literature (specifically, from references 60 and 62 in the references list).
8. Procedure: Page 5 line 190 add "to" they responded to...
Author’s response: We have added “to” to “they responded to” (see line 202).
9. Results: In the tables F1-IW, F2-BW - make clear what those codes are.
Author’s response: The codes F1, F2, … in the tables are the codes of the IPDMV and IBIPV dimensions. We have added this information (IPDMV dimensions, IBIPV dimensions) in the tables (see Table 2, Table 3, and Table 8).
10. Results: Check for commas vs periods in numbers. The y axes on most of the graphs have commas and they should be periods. There are commas in some of the chi square values as well.
Author’s response: We have checked all the text and Figures and we have changed the commas for periods in all the cases.
11. Results: 3.3 Other evidence section. Remind the reader here about this scale. I had to go back and look for what was being measured and on what response scale.
Author’s response: We have added an explanation about the scale (what was measured and what was the response scale) (see lines 292-296).
12. Results 3.4 Gender differences: I thought you recruited by sex not gender... be clear and consistent about this throughout the manuscript.
Author’s response: Gender (how social and cultural aspects can change attitudes towards IPVAW) is an important point to our research (not sex or biological aspects). That is why we ask about the participant's gender, by answering the question "Tell me what gender you identify with?" (We have added this assessment form, see line 190), and then we analyze the results in this sense (social and cultural aspects related to the fact that being men or women). Unfortunately we have not identify differences by gender among the age groups.
Reviewer 2 Report
The paper proposes an interesting survey of the Spanish population of favorable attitudes towards violence by the partner against women throughout the life cycle, filling a gap in the literature that focuses these studies mainly on the student population. The results show a U-shaped distribution: lower acceptance among middle-aged adults and young adults and higher acceptance among adolescents and older adults. However, when these attitudes were measured implicitly, IPVAW rejection increased with age, underlining how the age effect differs according to the measure of attitudes used and highlighting some difficulties related to measures based on reaction time among older people. .
Introduction
As the study is conducted on the Spanish population, it would be appropriate to broaden the description of the spanish context, both through the data of the incidence of IPV in the Spanish population, and by presenting previous studies on this population. In the Conclusions section there is a historical description of the investigation context which could also be also argued in Introduction section.
Participants
The recruitment procedure should be included in this section, which instead is included in the Procedures section.
Discussion
Gender differences are not sufficiently discussed in this section. It would be also interesting analyze intra-group gender differences, where a homogeneous distribution related to gender has been achieved. The limits of the research are not sufficiently discussed, for example I think it is a limitation that the socio-cultural differences of the sample are not considered as factors to analize.
Author Response
Responses to Reviewer 2:
Thank you very much for your assessment. We have made all the changes suggested by you.
Reviewer 2: Introduction. As the study is conducted on the Spanish population, it would be appropriate to broaden the description of the spanish context, both through the data of the incidence of IPV in the Spanish population, and by presenting previous studies on this population. In the Conclusions section there is a historical description of the investigation context which could also be also argued in Introduction section.
Author’s response: We have added some information about the Spanish context, specifically about the incidence of IPVAW and about attitudes of Spanish population towards IPVAW and their changes thorough the last decades (see lines 54-60). A description of the main results of previous studies in Spain on attitudes towards IPVAW and age is included on pages 2 and 3.
Reviewer 2: Participants. The recruitment procedure should be included in this section, which instead is included in the Procedures section.
Author’s response: We have included the recruitment procedure in the participants section as suggest the reviewer (see line 142-144).
Reviewer 2: Discussion. Gender differences are not sufficiently discussed in this section. It would be also interesting analyze intra-group gender differences, where a homogeneous distribution related to gender has been achieved. The limits of the research are not sufficiently discussed, for example I think it is a limitation that the socio-cultural differences of the sample are not considered as factors to analize.
Author’s response: The results obtained showed no differences by gender between age groups. This is why the discussion is centered in the analysis of age effect. We agree with the reviewer that maybe some more analysis may be made (such as analyzing intra-group gender differences or included socio-cultural differences) and we have included their absence as a limitation of our paper (see lines 469-474).